# High-Fat Diet Induces Pre-Diabetes and Distinct Sex-Specific Metabolic Alterations in Negr1-Deficient Mice

**DOI:** 10.3390/biomedicines9091148

**Published:** 2021-09-03

**Authors:** Maria Kaare, Kaie Mikheim, Kersti Lilleväli, Kalle Kilk, Toomas Jagomäe, Este Leidmaa, Maria Piirsalu, Rando Porosk, Katyayani Singh, Riin Reimets, Egon Taalberg, Michael K. E. Schäfer, Mario Plaas, Eero Vasar, Mari-Anne Philips

**Affiliations:** 1Institute of Biomedicine and Translational Medicine, Department of Physiology, University of Tartu, 19 Ravila Street, 50411 Tartu, Estonia; kaie.mikheim@ut.ee (K.M.); kersti.lillevali@ut.ee (K.L.); toomas.jagomae@ut.ee (T.J.); maria.piirsalu@ut.ee (M.P.); singhkat@ut.ee (K.S.); eero.vasar@ut.ee (E.V.); marian@ut.ee (M.-A.P.); 2Center of Excellence in Genomics and Translational Medicine, University of Tartu, 50411 Tartu, Estonia; kmck@ut.ee (K.K.); rando.porosk@ut.ee (R.P.); egon.taalberg@ut.ee (E.T.); 3Institute of Biomedicine and Translational Medicine, Department of Biochemistry, University of Tartu, 19 Ravila Street, 50411 Tartu, Estonia; 4Institute of Biomedicine and Translational Medicine, Laboratory Animal Center, University of Tartu, 14B Ravila Street, 50411 Tartu, Estonia; riin.reimets@ut.ee (R.R.); mario.plaas@ut.ee (M.P.); 5Institute of Molecular Psychiatry, Medical Faculty, University of Bonn, 53129 Bonn, Germany; este.leidmaa@uni-bonn.de; 6Department of Anesthesiology, Focus Program Translational Neurosciences, Research Center for Immunotherapy, University Medical Center of the Johannes Gutenberg-University Mainz, 55131 Mainz, Germany; Michael.Schaefer@unimedizin-mainz.de

**Keywords:** *Negr1*, obesity, metabolic disease, metabolomics, glucose intolerance, genetic models

## Abstract

In the large GWAS studies, *NEGR1* gene has been one of the most significant gene loci for body mass phenotype. The purpose of the current study was to clarify the role of NEGR1 in the maintenance of systemic metabolism, including glucose homeostasis, by using both male and female *Negr1*^−/−^ mice receiving a standard or high fat diet (HFD). We found that 6 weeks of HFD leads to higher levels of blood glucose in *Negr1*^−/−^ mice. In the glucose tolerance test, HFD induced phenotype difference only in male mice; *Negr1*^−/−^ male mice displayed altered glucose tolerance, accompanied with upregulation of circulatory branched-chain amino acids (BCAA). The general metabolomic profile indicates that *Negr1*^−/−^ mice are biased towards glyconeogenesis, fatty acid synthesis, and higher protein catabolism, all of which are amplified by HFD. *Negr1* deficiency appears to induce alterations in the efficiency of energy storage; reduced food intake could be an attempt to compensate for the metabolic challenge present in the *Negr1*^−/−^ males, particularly during the HFD exposure. Our results suggest that the presence of functional *Negr1* allows male mice to consume more HFD and prevents the development of glucose intolerance, liver steatosis, and excessive weight gain.

## 1. Introduction

Neuronal growth regulator 1 (*NEGR1*) is a candidate gene regulating human obesity, which encodes a neural cell adhesion and growth protein. *NEGR1* was identified as a member of the IgLON superfamily of neural cell adhesion molecules (CAMs), which also include LSAMP, NTM, OPCML, and IGLON5. GPI-anchored IgLONs have been shown to promote and guide neurite growth [1,2,3,4] and act as structural elements in the synapse that stabilize pre- and postsynaptic sides [5].

In GWAS studies, *NEGR1* gene locus has been repeatedly shown to have strong associations with human body mass index (BMI), indicating a role in body weight regulation and obesity [6,7,8,9,10,11]. Besides SNP markers, it has been found that two deletions (43 kb and 8 kb) upstream of *NEGR1* are strongly associated with early onset of extreme obesity [12].

Strong associations also appeared when genetic polymorphisms in the *NEGR1* gene were linked with dietary intake [13,14] and with psychological features generally associated with eating disorders [15].

In accordance with the initially described role of NEGR1 in the brain, polymorphisms in *NEGR1* have emerged among the most robust associations across different psychiatric disorders, including autism spectrum disorder, major depression, and schizophrenia [16]. The strongest GWAS associations, however, have linked *NEGR1* with depression [17,18]. The most recent data from depression patients suggest that the functional impact of NEGR1 might involve systemic regulation of homeostasis, as significant upregulation of NEGR1 has been shown in the hypothalamus [19] and peripheral blood of depression patients [20].

Due to an earlier described role in neuritogenesis, it has been speculated that *Negr1* also regulates neurite outgrowth in hypothalamic neurons [21,22]. Current evidence implies that Negr1 in the hypothalamic area decreases food intake. Higher levels of NEGR1 in hypothalamic nuclei were linked with lower food intake; administration of NEGR1 ectodomains into the paraventricular nucleus of the hypothalamus induced ~20% decrease in food intake in rats [23]. *Negr1* overexpression in the periventricular hypothalamic region, however, did not affect body weight or food intake, whereas knockdown of *Negr1* expression in the same nucleus increased body weight [24]. Exposure to a restricted feeding schedule has been shown to increase *Negr1* (22%) expression in the arcuate nucleus/ventromedial hypothalamus of rats [25]. Similarly, NEGR1 protein expression is increased in the lateral hypothalamus of fasted chicks [26], further suggesting the role of *Negr1* also as a regulator of a negative energy balance.

Accumulating evidence suggests non-central nervous system function of NEGR1 in intracellular lipid storage. Kim et al. [22] have demonstrated that NEGR1 interacts with cholesterol (CHOL) transporter 2 (NPC2), a key player in intracellular CHOL trafficking, and increases the stability of NPC2 in the late endosomes. Furthermore, Sandholt et al. [27] have shown that NEGR1 tag SNP (rs2568958) has significant associations with LDL cholesterol levels. In comparison to its most abundant expression in the brain, however, *Negr1* is only modestly expressed in the liver in humans and mice [28,29,30].

Among peripheral tissues, the expression of *Negr1* is high in the adipose tissue, remaining approximately four to five times lower than *Negr1* expression in the brain [29]. Bernhard et al. [31] detected lower NEGR1 expression in the subcutaneous adipose tissue (SAT) compared with visceral adipose tissue, whereas NEGR1 expression was lower in the SAT of obese humans compared to lean subjects. Walley et al. [29] demonstrated that, in the human SAT, NEGR1 appears to be central to the set of functionally related genes most differentially expressed between lean and obese subjects. An et al. [32] have shown that the level of lipid droplets (LD) was reduced in Negr1-overexpressing cells, whereas the intracellular LD level was higher in primary adipocytes obtained from *Negr1*^−/−^ mice than those from wild-type (WT) mice [32]. In addition, the expression level of LD-associated protein perilipin-2/ADRP increased in white adipose tissue of *Negr1*-deficient mice. Evidence from Bernhard et al. [29] suggests that NEGR1, which is upregulated during adipogenesis, is important in the adipocytes from early development. They also found that knockdown of NEGR1 significantly inhibited adipocyte maturation.

Data from mouse models with nonfunctional *Negr1* gene have revealed somewhat conflicting data. In the study of Lee et al. [21], both *Negr1* deficiency and loss-of-function mutation of *Negr1* resulted in a slight but steady decrease in body mass, whereas no change in body weight in *Negr1*^−/−^ mice was found by Joo et al. [33]. However, these works agree that lacking or nonfunctional NEGR1 protein causes alterations in the body composition, namely a decrease in muscle/lean mass. Joo et al. [33] showed a significant increase in fat mass with hypertrophic adipose cells containing enlarged cytosolic lipid droplets in *Negr1*^−/−^ mice compared with the WT mice. Moreover, these mice showed significant hepatic lipid accumulation, and a decrease in muscle mass and capacity [33].

In conclusion, accumulating evidence suggests that the systemic effect of *Negr1* in the regulation of body weight phenotype might be mediated both by the ability of *Negr1* to promote the cell–cell adhesion and neuritogenesis in the hypothalamic area [21,23] and by the independent role of *Negr1* in the regulation of fat trafficking/accumulation in peripheral tissues. Walley et al. [29] have shown that there is a high correlation in the expression levels of *NEGR1* between human subcutaneous adipose and hypothalamic tissues, suggesting a linked function for *NEGR1* across tissues.

The purpose of the present study was to shed light on how *Negr1* is implicated in food intake and systemic metabolism. We studied *Negr1*^−/−^ mice from both sexes receiving standard and high-fat diets. We also aimed to provide initial comparative information about liver steatosis [33] and reduced muscle mass [21,33] that have been described earlier in independently created *Negr1*-deficient mouse strains to further validate the phenotype. We confirmed that *Negr1*-deficient mice are prone to liver steatosis and male mice have signs of muscle atrophy, even when receiving a normal diet. At the same time, we showed that sex-specific metabolic alterations, including glucose intolerance phenotype in the *Negr1*^−/−^ mice, appeared only in mice receiving a high-fat diet, emphasizing the importance of sex and the interaction of genes and environment in maintaining homeostasis.

## 2. Materials and Methods

### 2.1. Animals

Male and female wild-type (WT) mice and their homozygous *Negr1*-deficient littermates (*Negr1*^−/−^), described previously [21] in F2 background ((129S5/SvEvBrd × C57BL/6N) × (129S5/SvEvBrd × C57BL/6N)), were used in the present study. Mice were group-housed in standard laboratory cages measuring 42.5 (L) × 26.6 (W) × 15.5 (H) cm, with 10 animals per cage in the animal colony, at 22 ± 1 °C under a 12:12 h light/dark cycle (lights off at 19:00 h). A 2 cm layer of aspen bedding (Tapvei, Estonia) and 0.5 l of aspen nesting material (Tapvei, Estonia) were used in each cage and changed every week. Water and food pellets (R70, Lactamin AB, Sweden) were available ad libitum. Breeding and the maintenance of the mice were performed at the animal facility of the Institute of Biomedicine and Translational Medicine, University of Tartu, Estonia. The use of mice was conducted in accordance with the regulations and guidelines approved by the Laboratory Animal Centre at the Institute of Biomedicine and Translational Medicine, University of Tartu, Estonia. All animal procedures were conducted in accordance with the European Communities Directive (2010/63/EU) with permit (No. 150, 27 September 2019) from the Estonian National Board of Animal Experiments.

### 2.2. Diet Composition

High-fat (HF) chow (DIO-45 kJ% fat (lard)) corresponds to the D12451 diet from Ssniff Spezialiäten GmbH (Soest, Germany) and its physiological energetic value is 4.615 kcal/kg. It contains 45 kJ% fat, 20 kJ% proteins, and 35 kJ% carbohydrates. This diet is characterized by high fat content (lard) and high sucrose levels. It is used to induce obesity and metabolic syndrome/diabetes in rats and mice.

The caloric value of regular chow (V1534-000 rat/mice universal maintenance diet, autoclavable (10mm) from Ssniff Spezialiäten GmbH, Soest, Germany) corresponds to 3.225 kcal/kg. It contains 9 kJ% fat, 24 kJ% proteins, and 67 kJ% carbohydrates. This diet is suitable for long-term experiments. 

### 2.3. High Fat Diet

The chronic high-fat diet experiment was performed with two different batches of mice (Figure 1). For the first batch of mice, 19 (±1)-week-old WT and *Negr1*^−/−^ male and female mice were divided into two groups; one group received regular food and another group received an HF diet (Ssniff Spezialiäten, Soest, Germany) (12 males and 12 females WT mice + 12 males and 12 females *Negr1*^−/−^ mice). Mice in the first batch received the HF diet (Ssniff Spezialiäten, Soest, Germany) for 7 weeks. All the mice were weighed weekly, starting from 10 weeks before the beginning of the HF diet (at the age of 10± 1 weeks); altogether, the body weight dynamics of the mice were tracked for 16 weeks. In the second batch of mice, 15 (±1)-week-old WT and *Negr1*^−/−^ male and female mice were divided into two groups; one group received regular food and another group the HF diet (Ssniff Spezialiäten, Soest, Germany) (10 males and 10 females WT mice + 10 males and 10 females *Negr1*^−/−^ mice). In the second batch, all mice received the HF diet (Ssniff Spezialiäten, Soest, Germany) for 6 weeks. As female mice showed no genotype effect in the glucose tolerance test, females received the HF diet (Ssniff Spezialiäten, Soest, Germany) for another 7 weeks (total of 13 weeks for females). In the second batch of mice, the food consumed was also weighed to evaluate their food consumption. At the end of both experimental periods, brain tissue, liver, and plasma were collected from all mice.

### 2.4. Food Preference Test (Batch III)

The food preference test was performed with minor modifications as described earlier in Leidmaa et al. [34]. All the mice were housed in single cages and, on days 1–3, mice received regular food. Both food and mice were weighed every morning at the same time (8.00 a.m.). On the morning of the 4th day, the mice and food (Ssniff Spezialiäten, Soest, Germany) were weighed, and the food was further weighed at specific timepoints (1 h, 3 h, 6 h, and 24 h) throughout the day. On the 24 h timepoint the next morning (5th day), mice were weighed, and the HF food (Ssniff Spezialiäten, Soest, Germany) was added for the food preference experiment. Both regular food and HF food were weighed at the same timepoints (1 h, 3 h, 6 h and 24 h) as the previous day. On the 24 h weighing (6th day), mice were weighed again, and HF food was removed; only regular food was retained. On the 7th day, the food was weighed for the last time to see if *Negr1*^−/−^ mice showed any withdrawal effects. 

### 2.5. Sample Collection

Mice were euthanized by decapitation, and trunk blood was collected into EDTA-coated microcentrifuge tubes and stored at 4 °C. All the tubes were centrifuged at 2000× *g* for 15 min at 4 °C. Plasma supernatant was separated and stored at −80 °C until further analysis. 

### 2.6. Measurement of Metabolites

AbsoluteIDQ™ p180 kit (BIOCRATES Life Sciences AG, Innsbruck, Austria) was used to determine plasma levels of metabolites according to the manufacturer’s protocol. Amino acids and biogenic amines in the samples were measured using the liquid chromatography–mass spectrometry techniques. Acylcarnitines (Cx:y), hexoses, sphingolipids (SMx:y or SM (OH)x:y), glycerophospholipids (lysophosphatidylcholines (lysoPCx:y)), and phosphatidylcholines (PCaa x:y and PC ae x:y) were measured using flow injection mass spectrometry. For both modes of analyzing, multiple reaction monitoring was used. Concentrations of the metabolites were calculated automatically by the MetIDQ™software (BIOCRATES Life Sciences AG, Innsbruck, Austria) in μM. The analytical system was QTRAP 4500 (Sciex, Framingham, MA, USA) in combination with Agilent 1260 series high-performance liquid chromatography (HPLC) (Agilent Technologies, Waldbronn, Germany).

Citric acid cycle intermediates were analyzed on the same instrument with an in-house protocol. In total, 50 µL serum was treated with 20 µL 100 µM [2H4] succinate (internal standard) and 750 µL ice-cold methanol for 10 min for protein precipitation. After centrifugation for 10 min at 21,000× *g*, the supernatant was dried under a stream of nitrogen and dissolved in 100 µL of methanol with 0.2% formic acid. The multiple reaction monitoring transitions in negative ionization mode were as follows: malate 133/115, succinate 117/73, citrate 191/87, pyruvate 87/43, alpha-ketoglutarate 145/101, lactate 89/43, oxaloacetate 131/87, beta-hydroxybutyrate 103/59, and the internal standard 121/77. 

### 2.7. Glucose Tolerance Test (GTT, Batch II)

Animals were deprived of food for 3 h before and during the experiment; water was available throughout the experiment. After measuring the basal glucose levels, the mice were intraperitoneally administered a glucose (Sigma-Aldrich, Burlington, MA, USA) solution in 0.9% saline (20% w/vol) at a dose of 2 g/kg of body weight. Blood glucose values were subsequently measured after 30, 60, 90, 120, and 180 min from the tail vein using a hand-held glucometer (Accu-Check Go, Roche, Mannheim, Germany). The bioavailability of glucose was estimated by calculating the under-the-curve area of plasma concentration at measured timepoints (AUC). For males, GTT was performed on the 6th week, and, for females, on 6th and on the 13th week of a high-fat diet.

### 2.8. Neutral Lipid and Actin Staining on the Tissue Cryosections

Male mice (n = 5 per group) from batch II were euthanized by decapitation, and dissected quadriceps femoris muscle and left lobe of the liver were immersed in 2-methylbutane (ACROS Organics™, Cat# 10511754, Carlsbad, CA, USA), precooled to −40 °C, and kept at −80 °C until further use. Cryosections (15 μm) were prepared using Leica low-profile disposable blades (DB80LS, Leica, Wetzlar, Germany) mounted to Leica CM1850-Cryostat (Leica, Wetzlar, Germany). Sections were collected onto Thermo Scientific™ SuperFrost Plus^TM^ slides (Thermo Scientific, Cat# 10149870) and kept at −80 °C. In order to minimize experimental errors caused by washes, staining incubations, etc., tissues from animals of each experimental group were collected on the same slide (four animals per slide).

Sections were immersion-fixed in 4% paraformaldehyde (PFA, Acros Organics™, Cat# 11924801, USA)/PBS for 15 min at 37 °C in a Coplin jar and washed thrice in PBS for 5 min each. Subsequently, sections were incubated with Alexa Fluor^®^ 555 Phalloidin (1:500, Invitrogen, Cat# A34055, Waltham, MA, USA) and BODIPY 493/503 dye (1 µM, Invitrogen, Cat# D3922, Waltham, MA, USA) in PBS over 10 min at 37 °C in a Coplin jar protected from light. Staining solution was obtained by diluting 5 mM BODIPY 493/503 in DMSO (Sigma Aldrich, Cat# D8418, Burlington, MA, USA) stock. Sections were washed thrice with PBS over 5 min each and stained with Hoechst 33,258 (5 μg/mL, Invitrogen, Cat# H1398, Waltham, MA, USA) in PBS over 5 min. Sections were subsequently rinsed with ddH_2_O and mounted as described above. To visualize filamentous actin fibers, sections were immersed in phalloidin conjugates. Images were acquired with a DP71 CCD camera (Olympus, Tokyo, Japan) mounted on a BX51 microscope (Olympus, Japan). Morphometric measurements of muscle fibers were performed manually using ImageJ software version 1.53c [35], and at least 70 muscle fibers were measured for each mouse. 

### 2.9. Statistical Analysis

Results are expressed as mean values ± SEM. Statistical analyses for metabolomic data, body weight, GTT, and food preference test were performed using GraphPad Prism 6 software (GraphPad, San Diego, CA, USA). Normal distribution of data was evaluated by the Shapiro–Wilk test. Comparison of metabolomic data between groups was performed using two-way ANOVA (diet × genotype), followed by a Bonferroni post hoc test. Comparison of GTT data between groups was performed using two-way ANOVA, followed by a Tukey post hoc test. Statistical analysis of the food preference test and cross-sectional area of muscle fibers was performed by using Mann–Whitney U-test. All differences were considered statistically significant at *p* < 0.05.

## 3. Results

### 3.1. Negr1 Deficiency Induces Lower Intake of HF Food but Higher Body Weight Gain in Male Mice

In the first batch of mice, the mice were weighed weekly from age 10 ± 1 weeks during the 9 weeks before the beginning of an HF diet; altogether, the body weight was tracked for 16 weeks. The variation in body weight was relatively high; therefore, no statistical genotype differences in body weight dynamics were detectable in the current study. In general, both male and female *Negr1*^−/−^ mice tended to have slightly lower body weight when on a standard diet (Figure 2b). When consuming a high-fat (HF) diet, however, male *Negr1*^−/−^ mice tended to gain more body weight compared to their WT littermates (Figure 2a). When mice consumed regular chow, there were no significant weight differences between genotypes (Figure 2b). Male *Negr1*^−/−^ mice consumed less HF food in the short-term food preference test, in which the food was individually measured for 24 h (Figure 2d). Correspondingly, a tendency for a lower HF food intake in male *Negr1*^−/−^ mice was also observed in group-housing settings 2 weeks before the glucose tolerance test (Figure 2e). Additionally, male *Negr1*^−/−^ mice also consumed smaller amounts of standard food when the consumption of food was individually measured for 96 h (Figure 2c). More detailed information about food intake on individual days and during the 1, 3, and 6 h food intake measurements have been shown in Appendix A. 

### 3.2. HFD Leads to Higher Levels of Blood Glucose in Negr1^−/−^ Mice, Whereas Phenotype Difference in Glucose Tolerance Test Was Apparent Only in Males

The HF diet elevated the level of basal blood glucose in both sexes; for males, there was a diet effect (*p* = 0.0029, F = 10.23), whereas females showed a genotype effect (*p* = 0.0001, F = 19.11). However, the basal level of glucose was statistically higher in the HF-diet-fed *Negr1*^−/−^ mice compared to the HF-diet-fed WT group (*p* = 0.0360) in both sexes (Figure 3a). The HF diet increased the basal level of blood glucose in *Negr1*^−/−^ male mice (*p* = 0.0117) (Figure 3b). In the female group, there was no diet effect (F = 0.01256); on the other hand, the genotype effect was observed (F = 19.11). The basal level of glucose was significantly higher in the HF-fed female *Negr1*^−/−^ group compared to the HF-diet-fed WT group (*p* = 0.0015) (Figure 3c).

For male mice, GTT was performed on the 6th week of the HF diet, and, for females, on the 6th and 13th week of the diet. In GTT, AUC was calculated for every mouse, and two-way ANOVA and Tukey post hoc tests were used.

In male mice, the AUC of HF-fed *Negr1*^−/−^ mice was statistically significantly higher compared to *Negr1*^−/−^ mice fed regular chow (*p* = 0.0008) (Figure 3d). In female mice, the AUC of the HF-diet-fed mice on the 6th week of diet was statistically significantly higher compared to the regular chow group in both WT (*p* = 0.0026) and *Negr1*^−/−^ (*p* = 0.018) mice (Figure 3e). In the 13th week, the results were similar to the 6th week; the AUC of female HF-diet-fed mice was statistically significantly higher than regular-chow fed mice in both genotypes: WT (*p* = 0.0015), *Negr1*^−/−^ (*p* = 0.0003) (Figure 3f). When the different timepoints were analyzed separately, the blood sugar levels of HF-diet-fed male mice were significantly higher compared to HF-diet-fed WT mice at 30 min (*p* = 0.0017) and 60 min (*p* = 0.0035) timepoints (Figure 3g). The blood sugar levels of the HF-diet-fed *Negr1*^−/−^ were significantly higher at 30 min (*p* < 0.0001), 60 min (*p* < 0.0001), 90 min (*p* = 0.0003), and 120 min (*p* = 0.0130) timepoints compared to regular-chow-fed *Negr1*^−/−^ mice (Figure 3g). In female mice, on the 6th week of diet, the blood sugar levels of the HF-diet-fed *Negr1*^−/−^ mice were significantly higher at 30 min (*p* = 0.0004) and 60 min (*p* = 0.0171) timepoints compared to regular-chow-fed *Negr1*^−/−^ mice, and, for the HF-diet-fed WT mice, the blood sugar levels were significantly higher at 30 min (*p* < 0.0001) and 60 min (*p* = 0.0003) timepoints (Figure 3h). On the 13th week of the diet, the blood sugar levels of the HF-diet-fed *Negr1*^−/−^ female mice were significantly higher at 30 min (*p* = 0.0116), 60 min (*p* = 0.0030), 90 min (*p* = 0.0002), and 120 min (*p* = 0.0049) timepoints compared to regular-chow-fed *Negr1*^−/−^ mice (Figure 3i). In the WT group the blood sugar levels of the HF-diet-fed mice were significantly higher at 30 min (*p* < 0.0001), 60 min (*p* = 0.0025), and 120 min (*p* = 0.0417) timepoints (Figure 3i).

For the batch I of mice, the level of hexoses was measured using the AbsoluteIDQ™ p180 kit. Although the profile of those results was slightly different from the results of basal glucose level, the HF diet also elevated the level of hexoses similarly to the basal level of glucoses. When the data of both sexes were pooled together, the HF diet significantly increased the level of hexoses (*p* = 0.027) in the *Negr1*^−/−^ group. If the sexes were analyzed separately, the increase in hexoses remained statistically significant only in the female group (Appendix A). In female mice, the HF diet increased the level of hexoses, both in the WT (*p* = 0.044) and *Negr1*^−/−^ (*p* = 0.018) groups (Appendix A).

### 3.3. HFD Induces an Altered Profile of Circulating Lipids Sex-Specifically in Negr1^−/−^ Mice

The level of saturated fatty acids (SFA) was markedly increased in the HF-diet-fed *Negr1*^−/−^ male mice group (*p* < 0.0001) (Figure 4a). The level of SFAs were statistically significantly higher in the HF-diet-fed *Negr1*^−/−^ group compared to the regular-chow-fed *Negr1*^−/−^ mice (*p* = 0.0173) and HF-diet-fed WT mice (*p* = 0.0086) (Figure 4a). In the female mice group, the HF diet increased the level of SFAs similarly for both genotypes (Figure 4b).

Serum levels of phosphatidylcholines (PC), acylcarnitines, and sphingomyelins (SM) were measured. Out of 100 quantifiable PC and SM species 87 were significantly altered due to HF diet. Among acylcarnitines, C18 and C18:1 were significantly increased by HF diet, but C0, C2, C3, C4, C14, and C18:2 decreased significantly. If C2 level was looked at in both sexes separately, there were genotype effects only for females (*p* = 0.0016) (Figure 4f). The level of C2 was statistically significantly lower in the regular-chow-fed *Negr1*^−/−^ female mice group compared to the regular-chow-fed WT female mice group (*p* = 0.0012) (Figure 4f).

There were not any statistically significant changes in the ratio of C2/C0 in male mice groups (Figure 4g). In female mice groups, the HF diet decreased the ratio of C2/C0 in the WT mice group (*p* < 0.0001) and, in the regular-chow-fed mice group, the ratio of C2/C0 was statistically lower in the *Negr1*^−/−^ group (*p* = 0.047) (Figure 4h). The HF diet increased the ratio of unsaturated PC/SFA in the WT group for both males (*p* = 0.0005) (Figure 4c) and females (*p* = 0.046) (Figure 4d). Although acylcarnitines with hydroxyacyl or dicarboxylic residues were frequently below the limit of quantification, or even below the limit of detection, their relative cumulative abundance among all acylcarnitines was higher in the HF diet (*p* < 0.0001).

### 3.4. HFD Induced an Increase in Circulating Amino Acids in Negr1^−/−^ Mice, More Prominently in Males

To identify the differences caused by the HF diet between WT and *Negr1*^−/−^, two-way ANOVA (diet (regular or HF) × genotype (WT or *Negr1*^−/−^)) and Bonferroni post hoc test were used. In WT animals, the HF diet had a limited effect on serum amino acid levels.

When both sexes were analyzed separately, the total level of amino acids was statistically significantly increased only in the HF-diet-fed *Negr1*^−/−^ male mice group (*p* = 0.023) (Figure 5a); in females, there were no statistically significant changes (Figure 5b). The level of branched-chain amino acids (BCAA) was also statistically significantly increased only in the HF-diet-fed *Negr1*^−/−^ male mice group (*p* = 0.05) (Figure 5g). If different BCAAs were looked at separately, the levels of Leu (*p* = 0.036) (Figure 5m) and Val (*p* = 0.031) were statistically significantly increased in the HF diet *Negr1*^−/−^ group. When we analyzed both sexes separately, the levels of Leu were significantly increased only in the HF-diet-fed *Negr1*^−/−^ male mice group (*p* = 0.0077) (Figure 5m). The HF diet increased the level of Val in both the WT (*p* = 0.023) and *Negr1*^−/−^ (*p* = 0.0093) male mice groups (Appendix A).

In males, the HF diet increased the level of His (*p* = 0.0028), Ser (*p* = 0.0377), Thr (*p* = 0.0003), Pro (*p* = 0.0058), Asn (*p* = 0.0084) (Appendix A), and Lys (*p* = 0.0059) (Figure 5k) in the *Negr1*^−/−^ group. The level of Ala was not statistically significantly increased by HF diet, but, in the *Negr1*^−/−^ group, it showed a tendency towards increase (*p* = 0.061) (Figure 5i). In the male group, there were genotype effects for Ser (*p* = 0.042), Arg (*p* = 0.037) (Appendix A), and Gly (*p* = 0.011) (Figure 5q), all three were statistically significantly higher in the HF-diet-fed *Negr1*^−/−^ mice group compared to the HF-diet-fed WT mice group. In male WT groups, the HF diet decreased the level of Gln (*p* = 0.020) (Appendix A) and Gly (*p* = 0.0010) (Figure 5q).

In females, HF diet increased the level of Ala (*p* = 0.0003) (Figure 5j), Lys (*p* = 0.0064) (Figure 5l), and Thr (*p* = 0.0200) (Appendix A) in the *Negr1*^−/−^ group. The level of Lys was also increased in the WT group (*p* = 0.036). In other amino acids, there were no statistically significant changes caused by the HF diet.

### 3.5. Altered Profile of Circulating Organic Acids in Negr1^−/−^ Mice

A few organic acids, including the citric acid cycle intermediates, were quantified in the blood serum in order to better identify the flux of metabolites (for the detailed information see Appendix A). Male *Negr1*^−/−^ mice had significantly higher beta-hydroxybutyrate (F = 13.1, *p* = 0.0012), lactate (F = 14.7, *p* = 0.0007), pyruvate (F = 4.9, *p* = 0.035), citrate (F = 5.5, *p* = 0.026), and oxaloacetate (F = 5.0, *p* = 0.034) than WT male animals. Female mice had a weak oxaloacetate decrease due to diet (F = 4.7, *p* = 0.04), but, other than that, all diet and genotypes were similar. When both genders were combined, citrate (F = 8.7, *p* = 0.005) and lactate (F = 4.3, *p* = 0.05) remained significantly elevated in *Negr1*^−/−^ animals. Lactate ratio to glucose was lowered by HF diet in female animals of both genotypes (F = 9.7, *p* = 0.005). In males, on the other hand, genotype had a significant effect, with *Negr1*^−/−^ having more lactate per glucose (F = 11.1, *p* = 0.003).

### 3.6. Negr1 Decifiency Induces Hepatic Fat Accumulation in Both Male and Female Mice and Reduced Skeletal Muscle Volume in Males

As abnormal hepatic fat accumulation has been previously demonstrated in alternatively created *Negr1*^−/−^ mice [33], we studied the liver cryosections of *Negr1*^−/−^ male mice by using BODIPY dye, which is a fluorescent conjugate of fatty acids. In mice receiving a standard diet, markedly higher fatty-acid-specific staining could be detected in the hepatocytes from the liver of *Negr1*^−/−^ mice (Figure 6b) compared to the hepatocytes from WT mice (Figure 6a). The hepatocytes from WT mice receiving a high-fat diet (Figure 6c) were similar to the hepatocytes from *Negr1*^−/−^ mice receiving standard food. High-fat food did not markedly change the appearance of the liver in the *Negr1*^−/−^ mice (Figure 6d). The stainings from all individual mice can be seen in Appendix A. The supplementary section (Appendix A) of the current study also provides histology results of a small population of middle-aged females (8–9 months old).

As skeletal muscle atrophy has been shown in an alternative *Negr1*^−/−^ mouse strain [33], we studied the muscle cross-sectional area of quadriceps femoris muscle in *Negr1*^−/−^ males in comparison with their WT littermates in both standard and high-fat diet groups. We confirmed significantly lower average muscle fiber size in the *Negr1*^−/−^ mice (Figure 7b) compared to that of WT mice on a standard diet (Figure 7a). Fat food diet did not induce significant difference in the average muscle fiber size in either the *Negr1*^−/−^ or WT mice (Figure 7d–f). There were no statistically significant changes in the average muscle fiber size between female *Negr1*^−/−^ and WT mice (Appendix A).

## 4. Discussion

In the large GWAS studies, NEGR1 gene has been one of the most significant gene loci for both body mass phenotype [7,8,9] and depression [17,18]. Depression and obesity are leading public health concerns worldwide. Shared genetic risk factors between depression and obesity have been reported, which could be mediated through shared etiological pathways, such as dysfunction of the hypothalamic–pituitary axis [36]. In the current study, we aimed to shed light on the pleiotropic nature of the *NEGR1* gene. Our purpose was to add evidence that would enable us to determine whether the impact of NEGR1 is established mainly through its function as a cell adhesion molecule in the hypothalamus, or whether it also has a distinct role in the systemic metabolism, which could, in turn, contribute to the etiology of psychiatric disorders.

In the current study, both male and female *Negr1*^−/−^ mice tended to have slightly lower body weight when on a standard diet, in accordance with an earlier study in *Negr1*^−/−^ mice [21]. A HF diet leads to body weight gain in both genotypes and both genders. Surprisingly, genotype difference occurred only in male mice. The wild-type (WT) males tended to gain less body weight when on HF diet compared to *Negr1*^−/−^ male mice; however, this effect was not related to higher amounts of consumed food. On the contrary, the WT males consumed higher amounts of HF food in the food preference test, in which the consumed food was individually measured for 24 h. Furthermore, a tendency to eat less of the HF food in male *Negr1*^−/−^ mice and, nevertheless, gain more weight was also observed in group-housing settings over 2 weeks. Male *Negr1*^−/−^ mice also consumed smaller amounts of standard food when measured individually for 96 h.

Our data indicate that *Negr1* deficiency induces alterations in the efficiency of energy storage; surprisingly, lower intake of HF food is accompanied with higher body weight gain in male *Negr1*^−/−^ mice. Previously, it has been shown that mice with NEGR1 loss-of-function mutation (Negr1-I87N mice) exhibited decreased food intake but normal energy expenditure, thus fostering the positive association between NEGR1 expression and obesity [21].

Interestingly, the initial approach to and consumption of the HF food during the first hours of the food preference test was not altered in *Negr1*^−/−^ mice (Appendix A). Measuring the acute consumption of energy-dense and palatable HF food allows us to estimate the changes in the reward processing (hedonic liking/wanting), as well as the acute homeostatic mechanisms that regulate the control of food intake [34]. *Negr1*^−/−^ mice seem to have a normal hedonic appetite and the corresponding satiety induction in the beginning of the food preference test. During a longer exposure to HF food (24 h), however, the *Negr1*^−/−^ males reduce their energy intake more than the WT. This could be explained by the impaired glucose tolerance and corresponding metabolic alterations in these mice. Reduced food intake could be an attempt to compensate for the metabolic challenge present in the *Negr1*^−/−^ males, particularly during the HF diet exposure. Previous studies have also shown that a restricted feeding schedule increases *Negr1* (22%) in the arcuate nucleus/ventromedial hypothalamus of rats [25], and that NEGR1 protein is increased in the lateral hypothalamus of fasted chicks [26]. NEGR1 in certain hypothalamic nuclei might, therefore, lead to increased appetite, which would be in line with our findings of decreased food consumption in *Negr1*^−/−^ mice. These effects may be site-specific, however, as administration of NEGR1 ectodomains into the paraventricular nucleus of the hypothalamus is shown to induce an opposite effect: a 20% decrease in food intake in rats [23].

The role of *NEGR1* in glucose homeostasis has been previously demonstrated in many instances. Schlauch et al. [11] have shown by genome-wide association studies (GWAS) that, besides obesity in the general population, *NEGR1* gene also associates with BMI in type 2 diabetes patients, with abnormal glucose levels and impaired fasting glucose. A direct impact on serum glucose levels has been demonstrated in *Negr1*^−/−^ mice with a >1.3-fold increase in serum glucose and insulin levels [33]. While the level of leptin was substantially higher, the level of insulin-sensitizing adipokine adiponectin was lower in the *Negr1*^−/−^ mice [33].

Our experiments in this study support the notion that, when on the standard diet, males and females from both genotypes have similar basal glucose levels. HF food, however, leads to higher levels of blood sugar in *Negr1*^−/−^ mice, as reported by earlier studies. In male mice, HF diet resulted in altered glucose tolerance only in *Negr1*^−/−^ mice. In females, HF diet altered glucose tolerance in both genotypes, and no genotype effect appeared after either 6 weeks or 13 weeks. Although males are more likely to develop insulin resistance and hyperglycemia in response to nutritional challenges [37], impaired glucose tolerance is more prevalent in women [38]. The gender dependence of glucose metabolism is in fact a complex outcome of many factors, such as muscle mass, muscle-to-fat ratio, and the nature of dysfunction in insulin signaling [38]. It should also be noted that the animals in our experiment did not develop diabetes, although the glucose tolerance test indicated a significant impairment in glucose homeostasis under some experimental conditions. With WT male mice having a slightly elevated basal glucose, and females performing worse in the glucose tolerance test, current results are in accordance with previous reports.

Our results thus far suggest that the presence of *Negr1* allows male mice to consume more HF food and hinders the development of glucose intolerance and excessive weight gain.

The HF diet is expected to overload fatty acid metabolism in one way or another. If ketogenic, HF would upregulate beta-oxidation, ketogenesis, and gluconeogenesis. If the HF diet has enough carbohydrates, a large portion of dietary fats would be stored as fat and none of the previously mentioned metabolic pathways need to be activated.

We observed nearly unanimous increases in serum PC and SM lipids due to the HF diet. Acylcarnitines at the same time decreased, with the exception of stearyl- and oleyl-carnitine, which increased. According to the manufacturer, palmitoyl, stearyl, and oleyl residues are the most dominant lipids in the HF diet formula. A high load of long-chain acyl residues increased various species of lipids, while the amount of free carnitine and short-chain acylcarnitines decreased due to activity and low substrate specificity of carnitine-acyl transferases. Hence, the pattern of changes due to the HF diet was as expected.

Hydroxylated acylcarnitines and acyl residues with two carboxylic acids could not be properly quantified in most samples. Therefore, the observed relative increase in the total amount of all hydroxylated and dicarboxylic acyl residues may be erroneous. Even more, an overload of acyl residues is expected to activate omega oxidation, which generates dicarboxylic acids and, thereby, alleviates the overload of lipid catabolism pathways. Ketone bodies appeared not to be increased by HF diet, and the individual acylcarnitines, as well as their ratios, did not imply that beta-oxidation and ketogenesis intensified because of our dietary intervention.

The fact that serum lipoproteins have gender-dependent reference values is common knowledge in clinical chemistry. Variations in male and female serum PC species, besides cholesterol and triglycerides, have also been shown previously [39]. Higher beta-hydroxybutyrate levels in females on a standard diet have been reported before as well. [40]. Thus, the gender differences in lipids found here are in accordance with previously published data. Somewhat surprisingly, the genotype effect on lipid profile appeared to be marginal in our experiment.

In WT animals, the diet had a limited effect on serum amino acid levels. In *Negr1*^−/−^ animals, particularly in males, the HF diet increased the total pool of amino acids in blood. It did not seem to be related to the essentiality of glucogenicity of the amino acids. The most significant was the increase for Lys, Thr, Ala, Ser, and His. With the reduced relative abundance of proteins in the HF diet, the increased level of essential amino acids (Lys, Thr, and His, in particular) implies increased protein breakdown in the body. Indeed, decreased muscle mass has been demonstrated in another strain of *Negr1*^−/−^ mice [33] and the same result was replicated in the current study, suggesting that *Negr1*^−/−^ mice might also be prone to the protein breakdown in the case of standard feeding.

If the increased protein breakdown is accompanied by increased amino acid usage in peripheral tissues, Ala and Gln should increase in serum because of shuttling of the amino group to the urea production in the liver. Female mice had a relative increase in Ala, but males, on the contrary, showed a weakly significant decrease in Gln. Additionally, the urea cycle intermediates (Arg, Cit, Orn) and their ratios did not indicate an overload of the urea cycle. The ratio of short-chain acylcarnitines and BCAA decreased with HF diet for both sexes, which indicates that, although BCAA levels increase, they are not catabolized into short-chain acyl radicals, which would be a necessary step in their oxidation.

The decrease in Glu and Gln on HF while nearly all other amino acids are either unchanged or increased might be a meaningful anomaly. Most logically, Glu could be transaminated to alpha-ketoglutarate, enter the tricarboxylic acid cycle, and be used for energy, either directly or indirectly via gluconeogenesis. Indeed, there was a tendency of reduced levels of alpha-ketoglutarate and oxaloacetate in animals consuming the HF diet.

Interestingly, Ala and Ser have elevated concentrations in *Negr1*^−/−^ on HF, although they are closely related to pyruvate and 3-phosphoglycerate, serving as potential substrates for gluconeogenesis. Ala generation in the periphery/muscles may simply exceed its use in the liver. This notion is supported by the fact that lactate, another gluconeogenesis substrate closely related to Ala and pyruvate, is not increased by the HF diet. However, comparing Ala and Gln levels and Ser- and alpha-ketoglutarate-related amino acids, there seems to be a preference to use carbons in alpha-ketoglutarate form rather than any gluconeogenic substrate. Alpha-ketoglutarate would be converted to oxaloacetate, which, besides being a starting point for gluconeogenesis, can be converted to aspartate by transamination. The latter is needed for the urea cycle and elimination of excessive nitrogen from increased amino acid catabolism. Gluconeogenesis from pyruvate also goes over oxaloacetate, but requires energy investment for pyruvate carboxylation, and may, therefore, be less economic.

In male *Negr1*^−/−^ animals, Leu and Val are increased. BCAAs are generally upregulated in glucose intolerance [41] and, particularly, Leu is known to regulate glucose and protein metabolism [42]. Whether the BCAA increase stems from protein breakdown and contributes to glucose intolerance or whether their higher level is maintained to counter peripheral glucose resistance cannot be answered from this study. Interestingly, high Leu should inhibit protein breakdown and enhance protein synthesis [43].

Altogether, *Negr1*^−/−^ mice, particularly males, break down proteins in the body in order to be able to use some amino acids, while others accumulate. This reprograming of metabolism does not seem to overload amino acid catabolic pathways, but causes reduced muscle fiber size phenotype. This reprograming also reduces metabolic flexibility and paves the way to glucose intolerance.

The citric acid cycle is a central mitochondrial pathway, which is closely related to catabolic and anabolic pathways of many biomolecules, including fatty acids, amino acids, and glucose. Above, we discussed how certain amino acids in *Negr1*^−/−^ mice are a more important gluconeogenic source than in WT mice. An important note and maybe the root of all metabolic alterations are the increased citrate levels in *Negr1*^−/−^ mice. High citrate is a signal for energy excess which activates fatty acid synthesizing enzymes allosterically [44,45]; on the other hand, citrate inhibits glycolytic enzymes. Thus, even on a normal diet, *Negr1*^−/−^ mice are biased towards fatty acid synthesis and have reduced glycolytic efficiency. Alpha-ketoglutarate, succinate, malate, and oxaloacetate did follow citrate’s pattern only in male mice, though. In *Negr1*^−/−^, a part of citrate is diverted away from the citrate cycle into lipid synthesis. The latter is intense in hepatocytes and might cause the fatty liver phenotype if upregulated.

A GWAS suggests that insulin suppresses NEGR1 in adipocytes and the synthetic glucocorticoid dexamethasone induces *NEGR1* expression [29]. Insulin is known to enhance fatty acid synthesis from citrate; whether it does so via NEGR1 or independently, we cannot answer, but our results are in good accordance with these findings. Another study on monozygotic twins discordant for type 2 diabetes found *NEGR1* being upregulated in adipocytes of type 2 diabetic twins [46]. Here, the insulin level is expected to be high, but adipocytes do not recognize it properly. Upregulation of *NEGR1* in relative insulin deficiency suggests that NEGR1 has a functional role in how insulin regulates the activities of enzymes of glycolysis, citric acid cycle, and/or lipogenesis. Even more, Joo et al. [33] have shown that *Negr1* deficiency induces abnormal fat deposition in various peripheral cells, especially fat and liver tissue cells.

Still, *Negr1*^−/−^ did not display increased levels of acyl residues that would be expected if high citrate diverges from glycolysis into lipid synthesis. As the synthesis also demands high amounts of ATP and reductive NADPH, it may be ineffective. Therefore, on a normal diet, the mice do not gain excessive fat or body weight, and display a normal serum lipid profile. If glycolysis is inhibited, more amino acids are catabolized for energy production, leading to a risk of muscle wasting. HF diet impairs glucose utilization even further [47]; thus, the *Negr1*^−/−^, who already have trouble with glucose usage, are forced to utilize amino acids as a source of energy (Figure 8). Metabolism of muscle proteins, which are the main protein reserve, is highly dependent on sex hormones.

Gender-specific effects were most notable in some amino acid levels, which were increased in *Negr1*^−/−^ animals on an HF diet, and glucose intolerance and body weight gain, in which male WT mice performed the best. Among the genders and genotypes, the WT males have the highest muscle mass, which helps to absorb and utilize glucose in its excess, and also more amino acids for gluconeogenesis to cope with time in glucose deficiency. The gluconeogenetic activity of *Negr1*^−/−^ mice might be higher than in WT and, therefore, they are less capable of coping with additional requirements from an unbalanced diet. High demand for amino acids and their higher availability in males than in females might cause the most pronounced gender-specific effects on amino acid levels due to the HF diet. The Supplementary Section (Appendix A) of the current study also provides histology results of a small population of middle-aged females (8–9-month-old females versus 5-month-old males). These images suggest that the reduced muscle fiber size phenotype might be sex-specific and present only in male *Negr1*^−/−^ animals. The steatosis-prone phenotype seems to be present in both sexes of *Negr1*-deficient mice. These findings support our theory that the lipid metabolism of *Negr1*-deficient mice is prone to lipid synthesis and accumulation. Altogether, the results of the current study emphasize that the future metabolic studies in *Negr1*-deficient mice should be performed comparatively in males and females.

## 5. Conclusions

In summary, *Negr1*^−/−^ mice have higher protein catabolism than WT mice. With higher usage of amino acids, the energy from carbohydrates and lipids can be diverted into reserves, e.g., formation of fat stores. Dependence of protein metabolism on sex is also a putative reason why the genotype effect is more pronounced in male mice. The HF diet does promote prediabetes and fat accumulation, which amplifies the metabolic pathways already activated by the absence of *Negr1*. Our data show that *Negr1* is one of the genetic factors that, together with other signaling molecules (e.g., sex hormones) and consumed diet, contributes to the balance of systemic metabolism, including glucose homeostasis.

## Figures and Tables

**Figure 1 biomedicines-09-01148-f001:**
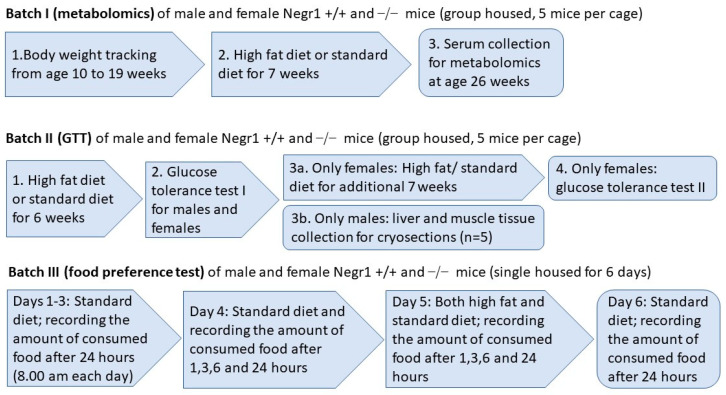
Schematic overview of the batches of mice and tests conducted in the current study. The age of Batch II and Batch III mice was around 12 weeks in the beginning of the experiments.

**Figure 2 biomedicines-09-01148-f002:**
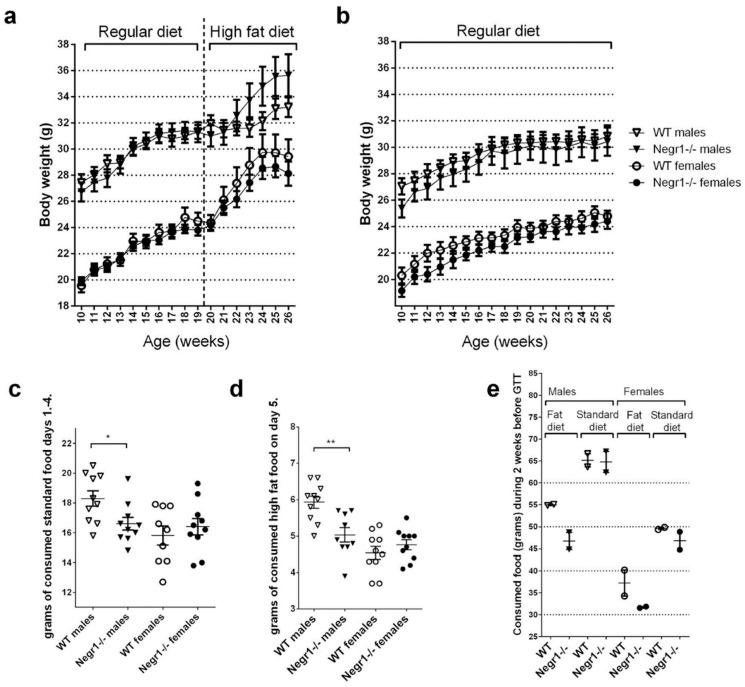
Body weight dynamics and food intake measurements. The body weight of the first batch of mice was measured 10 weeks before the beginning of the high-fat (HF) diet. Mice received the HF diet for 7 weeks. Body weight dynamics of (**a**) HF diet groups and (**b**) standard chow group. Daily food intake measurements in single-housed mice prior to the food preference test showed that (**c**) WT males consumed significantly more standard food (days 1–4). (**d**) WT males also consumed significantly more HF food during the food preference test (day 5) compared to the *Negr1*^−/−^ mice. (**e**) Food consumed (grams) for 2 weeks (14 days) before the glucose tolerance test in the second batch of mice. The data points were calculated as food consumed per group of mice (n = 5 per group). Data represent mean ± SEM. * *p* ≥ 0.05, ** *p* ≥ 0.01 (Mann–Whitney U-test).

**Figure 3 biomedicines-09-01148-f003:**
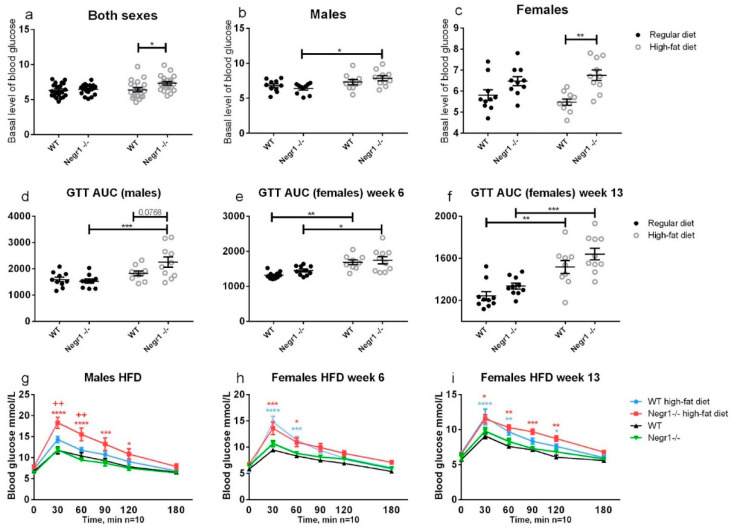
The basal level of glucose and glucose tolerance test. Basal level of glucose after 6 weeks of HF/standard diet when (**a**) both sexes were pooled together, (**b**) in the male group, and (**c**) in the female group. (**d**,**g**) Glucose tolerance test was performed for males after 6 weeks on the HF diet, and for females (**e**,**h**) after 6 and (**f**,**i**) 13 weeks of HF diet. (**d**–**f**) The bioavailability of glucose was estimated by calculating the area under the curve of plasma concentration (AUC) over the measured timepoints. (**g**–**i**) Mean values of blood sugar at different timepoints. Data represent mean ± SEM, * *p* ≥ 0.05, ** *p* ≥ 0.01, *** *p* ≥ 0.001, **** *p* ≥ 0.0001 (diet effect), ++ *p* ≥ 0.01 (genotype effect), two-way ANOVA (Bonferroni post hoc test (basal level of glucose), Tukey post hoc test (GTT)).

**Figure 4 biomedicines-09-01148-f004:**
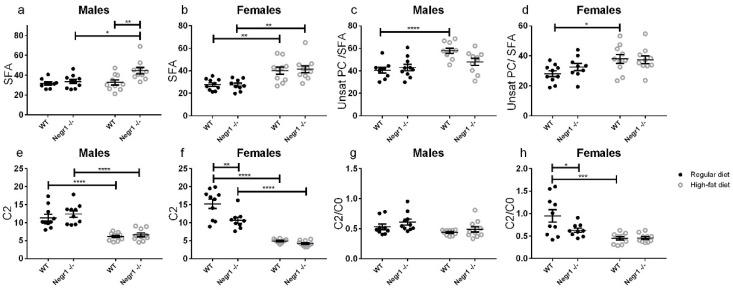
Effect of HF diet on the level of selected lipids and related ratios. The level of SFA for (**a**) males and (**b**) females, the ratio of unsaturated PC/SFA for (**c**) males and (**d**) females, the level of C2 for (**e**) males and (**f**) females, the ratio of C2/C0 for (**g**) males and (**h**) females. Data represent mean ± SEM, * *p* ≥ 0.05, ** *p* ≥ 0.01, *** *p* ≥ 0.001, **** *p* ≥ 0.0001, two-way ANOVA (Bonferroni post hoc test).

**Figure 5 biomedicines-09-01148-f005:**
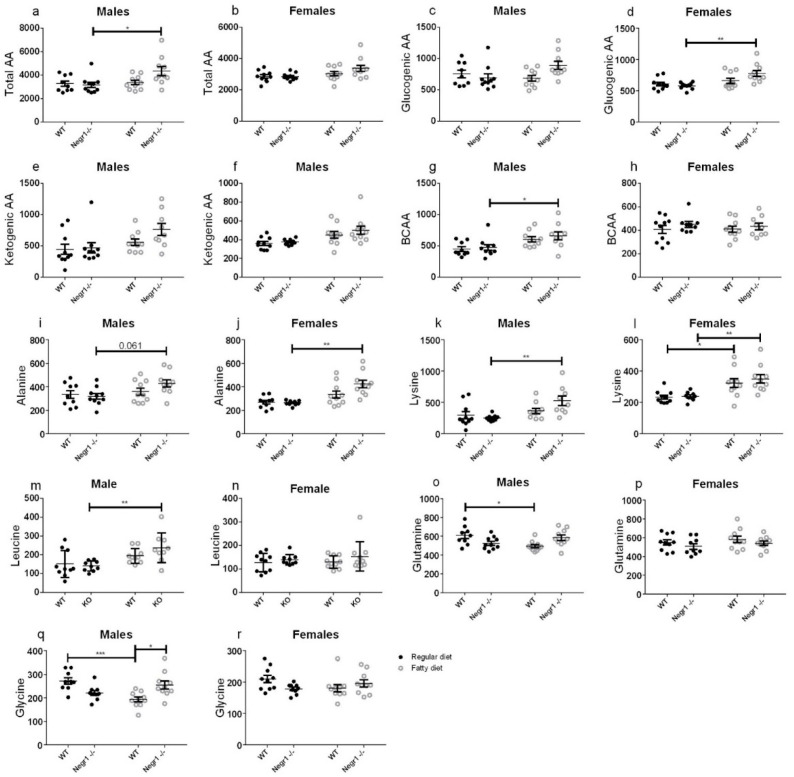
Effect of HF diet on the level of amino acids. In WT animals the HF diet had limited effect on serum amino acid levels. In *Negr1*^−/−^ animals the HF diet increased the total pool of amino acids in blood. The level of (**a**,**b**) Total AA, (**c**,**d**) Glucogenic AA, (**e**,**f**) Ketogenic AA and (**g**,**h**) BCAA. (**i**–**r**) The level of different amino acids. Data represent mean ± SEM, * *p* ≥ 0.05, ** *p* ≥ 0.01, *** *p* ≥ 0.001, two-way ANOVA (Bonferroni post hoc test).

**Figure 6 biomedicines-09-01148-f006:**
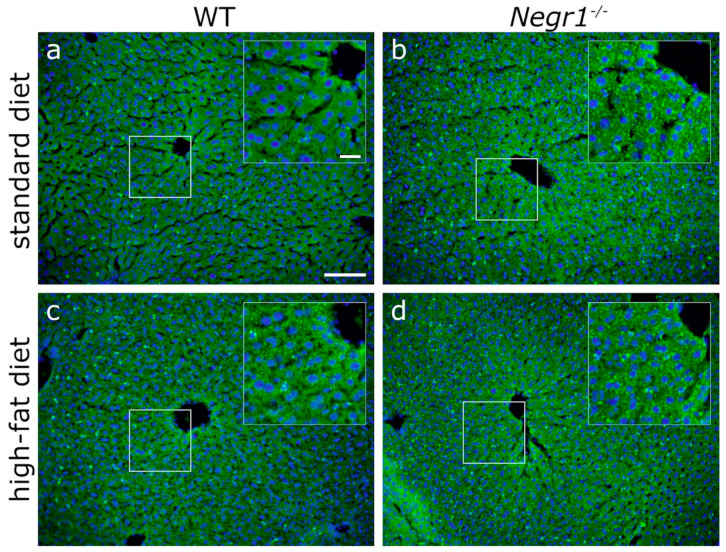
Hepatic lipid content. Representative images of BODIPY neutral lipid staining (green) from (**a**,**c**) WT and (**b**,**d**) *Negr1*^−/−^ mouse liver sections that underwent (**a**,**b**) standard and (**c**,**d**) high-fat diets. Compared to (**a**) WT mice, the hepatocytes from (**b**) *Negr1*^−/−^ mice in the standard diet display increased lipid droplet accumulation around the portal vein. This genotype-dependent difference is diminished with (**c**,**d**) high-fat diet treatment. Nuclei (blue) were stained using H33258 stain. Scale bars: 100 µm, 25 µm (inserts).

**Figure 7 biomedicines-09-01148-f007:**
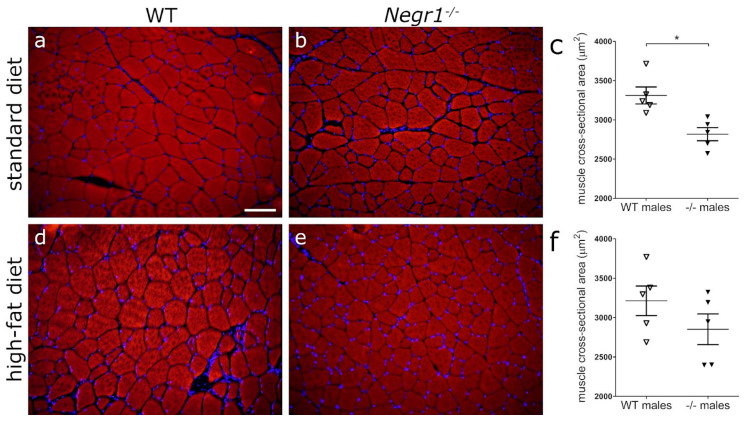
Reduced cross-sectional area of muscle fibers in *Negr1*^−/−^ mice. Phalloidin (red) stained cryosections from the quadriceps femoris muscle of *Negr1*^−/−^ (**b**,**e**) and wild-type (**a**,**d**) mice receiving standard (**a**,**b**) or high-fat (**d**,**e**) diets. (**c**) Morphometric measurements of muscle fiber cross-sectional area revealed significant decrease in *Negr1*^−/−^ mice in the standard diet group, * *p* ≥ 0.05 (Mann–Whitney U-test). (**f**) High-fat diet treatment resulted in no difference in muscle fiber cross-sectional area between genotypes. Nuclei (blue) were stained using H33258 stain. Scale bar: 100 µm.

**Figure 8 biomedicines-09-01148-f008:**
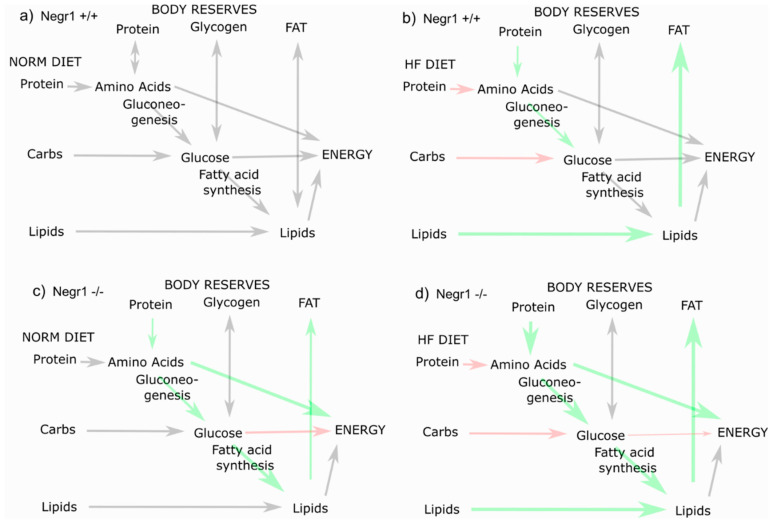
A hypothetical and simplified scheme of the metabolic differences in wild-type (*Negr1*^+/+^) (**a**,**b**) and *Negr1*^−/−^ (**c**,**d**) mice on a normal diet (**a**,**c**) and high-fat diet (**b**,**d**). Increased size and green color indicate increased metabolic flux, reduced arrow size and red color indicate inhibited process.

## Data Availability

The data presented in this study are available on request from the corresponding author.

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
