# Peer review of "High-Fat Diet Induces Pre-Diabetes and Distinct Sex-Specific Metabolic Alterations in Negr1-Deficient Mice"

_biomedicines, 2021, doi:10.3390/biomedicines9091148_

Round 1
Reviewer 1 Report
In this manuscript, Kaare and colleagues describe a role for NEGR1 in glucose homoestasis and energy storage, and assess the sex-specific role for NEGR1 in these effects. The authors fed standard diet or HFD and performed GTT, liver and muscle staining, and metabolomics, along with measuring food intake, in both male and female mice. The authors conclude from their findings that the absence of Negr1 promotes higher intake of HFD in male mice and leads to glucose intolerance and increased weight gain. While the data provided are of interest, some concerns exist with the interpretation and reporting of some of the data and detract from enthusiasm for the manuscript in its current form. These concerns are described below.
- There is no real mechanistic connection between the vast amount of metabolomics data and the phenotype observed. For example, the authors conclude that the KO mice have higher protein catabolism than WT mice but they do not provide any data that indicate how or why this may be occurring. The discussion is primarily based on what the data imply - could the authors provide some gene expression or functional data to support the metabolomics results?
2. The authors demonstrate that muscle fiber area is decreased in the quadriceps muscle of the male KO mice compared to WT and that this means they have decreased overall muscle mass. This is a conclusion that simply cannot be made based on one cross section of muscle. What about the muscle mass of the whole animal? Could the KO mice have larger muscles but smaller fibers? This must be addressed. Further, the study would be greatly strengthened by similar staining for females, since this is a manuscript about sex differences.
3. The liver staining is not convincing - Kupffer cell and hepatocyte markers should be used in addition to BODIPY and nuclear staining. Why is one of the KO livers in the supplemental figure full of fat while none of the other livers seem affected? Liver staining should be included for female mice.
4. The legend for Fig 4 is incomplete.
5. Can the authors comment on what they observed in adipose tissue from each of the sexes and genotypes? This is important if Negr1 KO has been demonstrated to promote adipocyte hypertrophy but prevent adipogenesis. This could also be important in terms of diet effects.
Author Response
Dear Editor, Ms. Paige Yu
Please find attached the revised manuscript titled “High fat diet induces prediabetes and distinct metabolic alterations sex-specifically in Negr1 deficient mice”. We thank you and the reviewers for their positive response to our manuscript and for constructive suggestions.
We have now revised the manuscript carefully to address the comments and concerns of the reviewers, and we feel that we have followed all of their recommendations as far as possible. These revisions have further strengthened the manuscript, and we hope that you will now consider it acceptable for publication in Biomedicines, section “Molecular and Translational Medicine” and as a part of a special issue “Emerging Paradigms in Insulin Resistance”. Please see the point-by-point response to the comments raised by the reviewers below. The respective changes have been highlighted by using the "Track Changes" function in a revised version of the manuscript.
Question/comment 1: There is no real mechanistic connection between the vast amount of metabolomics data and the phenotype observed. For example, the authors conclude that the KO mice have higher protein catabolism than WT mice, but they do not provide any data that indicate how or why this may be occurring. The discussion is primarily based on what the data imply - could the authors provide some gene expression or functional data to support the metabolomics results?
Answer 1: Thank you for your suggestions. In regard to genetic data, an analysis of how Negr1 knock-out influences other genes would be a nice addition, but the manuscript is already quite heavy on experimental data and interpretations. Linking gene expression with metabolite levels is not straightforward, as the metabolites (e.g. blood sugar) depend strongly on diet and environment. Furthermore, gene expression patterns are expected to be tissue/cell type specific, while metabolomic characterization is done on blood serum, whose composition depends more or less on all tissues. Thus, the proposed addition of a genomic or proteomic layer would significantly increase the complexity of data and would force us to set some other limitations in order not to overwhelm ourselves and the readers.
While we agree that functional genomics would be a good addition as a follow-up study, we find that the metabolism and phenotype connections have been discussed. Nevertheless, we have attempted to highlight them more and made the following changes to the manuscript. Following sentences are added to the article discussion section: 1) “. This reprogramming of metabolism does not seem to overload amino acid catabolic pathways but causes reduced muscle fiber size phenotype. This reprogramming also reduces metabolic flexibility and paves the way to glucose intolerance.” (Page 16 paragraph 2); 2) “The latter is intense in hepatocytes and might cause the fatty liver phenotype if upregulated” (page 16 paragraph 3).
Question/comment 2: The authors demonstrate that muscle fiber area is decreased in the quadriceps muscle of the male KO mice compared to WT and that this means they have decreased overall muscle mass. This is a conclusion that simply cannot be made based on one cross section of muscle. What about the muscle mass of the whole animal? Could the KO mice have larger muscles but smaller fibers? This must be addressed. Further, the study would be greatly strengthened by similar staining for females since this is a manuscript about sex differences.
Answer 2: We thank the Reviewer for pointing these issues out. The purpose of the histology section of the current study was to provide initial comparative information about overlapping phenotypes in two independently created Negr1-deficient mouse strains. Demonstration of muscle atrophy and liver steatosis in Negr1-deficient is not a novel finding but rather a validation of the phenotype. These changes have been demonstrated in two different Negr1-deficient mouse strains (Lee et al, 2012 and Joo et al, 2019). The initial nature of the histology section in the current study has now been emphasized in the last paragraph of “Introduction” by adding the following explanation: “We also aimed to provide initial comparative information about liver steatosis [33] and reduced muscle mass [21,33] that has been described earlier in independently created Negr1-deficient mouse strains to further validate the phenotype. We confirmed that Negr1-deficient mice are prone to liver steatosis and male mice have signs of muscle atrophy even when receiving a normal diet.”
We have now improved the description of the methodology, as at least 70 muscle fibers were measured for each mouse for assessing the average cross-sectional area of muscle fibers. The description has now been improved in the “Materials and methods'' section (subsection 2.8. Neutral lipid and actin staining on the tissue cryosections, page 6 paragraph 2).
The conclusion about decreased muscle mass was supported not only by the current results but also by previous studies. Lee et al. (2012) showed a reduction of approximately 8% in lean mass for the Negr1-187N mice in both females and males. Joo et al. (2019) demonstrated that Negr1-/- mice have muscle atrophy and impaired muscle function. The overlapping results from previous studies have now been more clearly emphasized in the “Introduction” section (page 2, last paragraph).
Similar muscle stainings for the females have now been added as Supplementary figure S8. Unfortunately, we did not perfectly have age-matched females available for the revision phase, and the histology has been done with female mice who are approximately 3 months older compared with males that were used for histology. Therefore, we have decided to keep the female stainings in the supplement as an additional observation that is generally supporting our study. The variation in the female group is too big to see the significant genotype differences but the data suggests that no prominent muscle atrophy, comparable with that of males, could be detected in females.
The following paragraph about the stainings of female mice has been added in the end of “Discussion” section (end of page 17).
“The supplementary section (S6 and S7) of the current study also provides histology results of a small population of middle-aged females (8-9 months old female versus 5 months old males). These images suggest that the reduced muscle fiber size phenotype might be sex-specific and present only in male Negr1−/− animals. The steatosis-prone phenotype seems to be present in both sexes of Negr1-deficient mice. These findings support our theory that the lipid metabolism of Negr1-deficient mice is prone to lipid synthesis and accumulation. Altogether the results of the current study emphasize that the future metabolic studies in Negr1-deficient mice should be performed compara-tively in males and females.”
Lee, A.W.S.; Hengstler, H.; Schwald, K.; Berriel-Diaz, M.; Loreth, D.; Kirch, M.; Kretz, O.; Haas, C.A.; Hrabe de Angelis, M.; Herzig, S.; et al. Functional Inactivation of the Genome-Wide Association Study Gene Neuronal Growth Regulator 1 in Mice Causes a Body Mass Phenotype. PLoS ONE 2012, 7, e41537. https://doi.org/10.1371/journal.pone.0041537
Joo, Y.; Kim, H.; Lee, S.; Lee, S. Neuronal growth regulator 1-deficient mice show increased adiposity and decreased muscle mass. Int J Obes. 2019, 43, 1769–1782. https://doi.org/10.1038/s41366-019-0376-2
Question/comment 3: The liver staining is not convincing - Kupffer cell and hepatocyte markers should be used in addition to BODIPY and nuclear staining. Why is one of the KO livers in the supplemental figure full of fat while none of the other livers seem affected? Liver staining should be included for female mice.
Answer 3: As we have written for the previous point, the purpose of the histology section of the current study was performed to provide initial validation of the phenotype.
The liver stainings for female mice have now been performed and added to the manuscript (Supplementary figure S6 and S7). In general, females are showing the same tendencies: Negr1 deficiency is related to the risk of steatosis, even in mice on normal diet.
The male mouse with the extremely strong steatosis had the highest body weight (Supplementary S5 q, 36.42 g) among Negr1 -/- mice on HFD, whereas the average body weight in this group was 32.05 ± 3.226 g (mean±SD). Therefore, our data from male mice indicates a link between the severity of steatosis and higher body weight. Some variety within a mouse group is expected as we use the mixed genetic background (C57Bl6/129Sv) and Negr1 -/- mutation might have variable penetrance across individual mice. Our experience and multiple other studies (Rivera and Tessarollo, 2008; McCutcheon et al, 2008; Xiao et al, 2017) suggest that a mixed background gives the advantage to better model the contribution of genetic effects in the genetically variable human populations. Groupwise, the phenotype effects can be clearly seen despite certain variance within the group.
As the male mouse with strongest steatosis and highest body weight (Supplementary S5 q) also had the strongest glucose intolerance (highest AUC value) in the glucose tolerance test, our preliminary data suggest a link between higher body weight, stronger steatosis, and more severely manifested glucose intolerance (Naeem et al. 2021). At the same time, we could see strong Negr1-deficiency induced glucose intolerance in another mouse with a lower body weight and less pronounced steatosis (Supplementary S5 r). Furthermore, the link between body weight and severity of steatosis seems not to be similar in female mice (Supplementary figure S6). Just for the initial information for the reader, the body weights of individual mice have now been marked to each individual image of liver staining (Supplementary S5 and S6). Further studies are needed to clarify the possibly sex-specific links between accumulation of fat in the liver and elsewhere in the body due to Negr1-deficiency and glucose homeostasis. At this time, however, we would not like to draw any conclusions from effects seen only in single individuals.
As the female mice we used for the BODIPY staining in the liver were approximately three months older (8-9 months old) than the males for the liver staining (5 months old), we find it best not present the data comparatively and the female data is shown in the supplementary section of the current study. The value of female images for the current study is that female images support the steatosis-phenotype of Negr1-deficient mice and also our general theory that the lipid metabolism of Negr1-deficient mice is prone to lipid synthesis and accumulation. As in females the steatosis seems to be more severe compared to males, it further emphasizes that the future metabolism related studies in Negr1-deficient mice need to be performed comparatively in both males and females. Our observations also open a possible importance of aging-related effects in Negr1-deficient mice.
We have now removed the claim that “wild-type mice liver undergoing a standard diet are observable mostly in Kupfer cells” from the figure caption of Supplementary figure S5. We agree that without specific markers, it is inappropriate to name specific cell types.
Rivera J., Tessarollo L. (2008). Genetic Background and the Dilemma of Translating Mouse Studies to Humans. Immunity, 28(1): 1 - 4. https://doi.org/10.1016/j.immuni.2007.12.008
McCutcheon J.E., Fisher A.S., Guzdar E., Wood S.A., Lightman S.L., Hunt S.P. (2008). Genetic background influences the behavioral and molecular consequences of neurokinin-1 receptor knockout. Eur J Neurosci. 27(3):683-90. https://pubmed.ncbi.nlm.nih.gov/18279320/
Xiao L., Sonne S.B., Feng Q., Chen N. et al. (2017). High-fat feeding rather than obesity drives taxonomical and functional changes in the gut microbiota in mice. Microbiome, 5:43. https://www.ncbi.nlm.nih.gov/pmc/articles/PMC5385073/
Lee, A.W.S.; Hengstler, H.; Schwald, K.; Berriel-Diaz, M.; Loreth, D.; Kirch, M.; Kretz, O.; Haas, C.A.; Hrabe de Angelis, M.; Herzig, S.; et al. Functional Inactivation of the Genome-Wide Association Study Gene Neuronal Growth Regulator 1 in Mice Causes a Body Mass Phenotype. PLoS ONE 2012, 7, e41537. https://doi.org/10.1371/journal.pone.0041537
Joo, Y.; Kim, H.; Lee, S.; Lee, S. Neuronal growth regulator 1-deficient mice show increased adiposity and decreased muscle mass. Int J Obes. 2019, 43, 1769–1782. https://doi.org/10.1038/s41366-019-0376-2
Naeem M., Bülow R, Schipf S., Werner N. et al. (2021). Association of hepatic steatosis derived from ultrasound and quantitative MRI with prediabetes in the general population. Scientific Reports, 11, 13276. https://doi.org/10.1038/s41598-021-92681-3
Question/comment 4: The legend for Fig 4 is incomplete.
Answer 4: Thank you for pointing this out. The legend for Fig 4. is now added to the article.
Question/comment 5: Can the authors comment on what they observed in adipose tissue from each of the sexes and genotypes? This is important if Negr1 KO has been demonstrated to promote adipocyte hypertrophy but prevent adipogenesis. This could also be important in terms of diet effects.
Answer 5: The adipocyte hypertrophy has been described in the alternatively created Negr1-deficient mouse line (Joo et al, 2019), whereas the sex of experimental animals was not specified; they used mixed genders. By including muscle and liver stainings from male mice, our purpose was the validation of the phenotype as these changes in the body composition have been now demonstrated in two different Negr1-deficient mouse strains. At the stage of tissue collection, we were not aware of the significant sex differences that would be evident in the metabolic dysfunction in Negr1-deficient mice as the current paper is the first to study and demonstrate sex-specific nature of the changes. We agree that studies focusing on metabolic changes, including studies of adipose tissue in Negr1-deficient mouse models should be done comparatively in both sexes in the light of the results of the current study. In the next steps, however, the contribution of the hormonal milieu to the fat tissue homeostasis in females should be studied in detail (Newell-Fugate, 2017), with careful consideration to the estrous cycle influences. This would be beyond the scope of the current study.
Newell-Fugate A.E. (2017). The role of sex steroids in white adipose tissue adipocyte function. Reproduction, 153: R133-R149. https://doi.org/10.1530/REP-16-0417
Joo, Y.; Kim, H.; Lee, S.; Lee, S. Neuronal growth regulator 1-deficient mice show increased adiposity and decreased muscle mass. Int J Obes. 2019, 43, 1769–1782. https://doi.org/10.1038/s41366-019-0376-2
Reviewer 2 Report
The study is interesting. it is conducted according to basic research standards. It elegantly shows the role of Negr1.It dissects the signaling pathways and proposes a simplified scheme of the metabolic differences. I have no comments to add.
Author Response
Dear Reviewer,
Please find attached the revised manuscript titled “High fat diet induces prediabetes and distinct metabolic alterations sex-specifically in Negr1 deficient mice”. We thank you for the positive response to our manuscript.
Yours sincerely,
Maria Kaare
Junior Scientist
Department of Physiology
Faculty of Medicine
University of Tartu
19 Ravila Street
Tartu 50411, Estonia
maria.kaare@ut.ee
Round 2
Reviewer 1 Report
The authors have adequately addressed the concerns of this reviewer.
Author Response
Dear reviwer,
I would like to thank You for your positive feedback to our article and for helping us to improve our manuscript.
Yours sincerely,
Maria Kaare
Junior Scientist
Department of Physiology
Faculty of Medicine
University of Tartu
19 Ravila Street
Tartu 50411, Estonia
maria.kaare@ut.ee